# Design and Optimization of a Radial Turbine to Be Used in a Rankine Cycle Operating with an OTEC System

**Khaled Alawadhi [1],\*, Yousef Alhouli [1], Ali Ashour [2] and Abdullah Alfalah [3]**

[1]  Department of Automotive and Marine Engineering Technology, College of Technological Studies,
     The Public Authority for Applied Education and Training, Shuwaikh, Kuwait City 70654, Kuwait;
     Ym.alhouli@paaet.edu.kw
[2]  Technical Department, Coast Guard, Ministry of Interior, Kuwait City 70654, Kuwait; Ahashour@moi.gov.kw
[3]  Automotive Department, Industrial Institute at Sabah Alsalem, The Public Authority for Applied Education
     and Training, Sabah Alsalem, Kuwait City 70654, Kuwait; alfalah@gmail.com
\*  Correspondence: ka.alawadhi@paaet.edu.kw

**Abstract:** Design and optimization of a radial turbine for a Rankine cycle were accomplished ensuring higher thermal efficiency of the system despite the low turbine inlet temperature. A turbine design code (TDC) based on the meanline design methodology was developed to construct the base design of the turbine rotor. Best design practices for the base design were discussed and adopted to initiate a robust optimization procedure. The baseline design was optimized using the response surface methodology and by coupling it with the genetic algorithm. The design variables considered for the study are rotational speed, total to static speed ratio, hub radius ratio, shroud radius ration, and number of blades. Various designs of the turbine were constructed based on the Central Composite Design (CCD) while performance variables were computed using the in-house turbine design code (TDC) in the MATLAB environment. The TDC can access the properties of the working fluid through a subroutine that links NIST's REFPROP to the design code through a subroutine. The finalization of the geometry was made through an iterative process between 3D-Reynolds-Averaged Navier-Stokes (RANS) simulations and the one-dimensional optimization procedure. 3D RANS simulations were also conducted to analyze the optimized geometry of the turbine rotor for off-design conditions. For computational fluid dynamics (CFD) simulation, a commercial code ANSYS-CFX was employed. 3D geometry was constructed using ASYS Bladegen while structured mesh was generated using ANSYS Turbogrid. Fluid properties were supplied to the CFD solver through a real gas property (RGP) file that was constructed in MATLAB by linking it to REFPROP. Computed results show that an initial good design can reduce the time and computational efforts necessary to reach an optimal design successfully. Furthermore, it can be inferred from the CFD calculation that Response Surface Methodology (RSM) employing CFD as a model evaluation tool can be highly effective for the design and optimization of turbomachinery.

**Keywords:** radial turbine design; optimization; organic Rankine cycle; response surface methodology (RSM)

## 1. Introduction

With ever-rising global energy demands, increasing apprehension about environmental issues is of concern, which include global warming, ozone layer depletion, and air pollution. This is causing the need for efficient and greener energy resources. One of the available renewable energy resources is ocean thermal energy conversion (OTEC). OTEC is based on the fact that there is a sufficient temperature gradient available between deep seawater and sea surface water, although the efficiency

of OTEC is low based on the available low-temperature gradient [1]. However, research in this field is progressing to improve cycle efficiency. Wang et al. [2] reported the significance of some important factors that could improve the efficiency of the cycle, i.e., evaporating pressure, working fluid, turbine exit pressure, and superheating temperature. Bharathan [3] conducted a study on OTEC to determine the possible prospects for the cycle to improve its effectiveness. Goto et al. [4] conducted a simulation study to optimize the Uehara cycle for an ammonia–water mixture as a working fluid. Yang and Yeh [5] performed a cycle optimization study on the OTEC cycle using different working fluids. Kim et al. [6] examined the performance of a dual-use OTEC open cycle. They proposed that steam mass fractions are an important factor to maximize production. Idrus [7] proposed a new Geo-ocean thermal energy conversion (GeOTEC) cycle that used the waste heat recovery from an offshore gas plant and ocean thermal energy in a single cycle. Goto et al. [8] developed a web application for the simulation of OTEC by employing a double Rankine cycle. Apart from improving the cycle efficiency through cycle optimization, the efficiency of the cycle could be enhanced by improving the efficiency at the component level. By increasing the efficiency of the turbine by 2%, cycle efficiency could be improved by 1% [9]. Thus, the current study focused on the design and optimization of a radial turbine of an OTEC system. A methodology for the design of a small-scale radial turbine for the organic Rankine cycle (ORC) has been suggested by many researchers but an optimization procedure does not appear in the literature. Li and Ren [10] developed a methodology of a small-scale radial turbine that is based on the real gas model. Moreover, they have predicted the aerodynamic performance of the turbine numerically. Zhai et al. [11] proposed a methodology for turbine design for an improved model of low-grade organic Rankine cycle. Mounier et al. [12] updated specific speed to specific diameter graphs for small scale radial turbines. They also performed a sensitivity analysis of different parameters associated with turbine geometry on its aerodynamic performance. Saeed and Kim [13,14] used a turbine meanline design code for the optimization of the centrifugal turbine and a compressor for a supercritical Carbone dioxide Brayton cycle.

Centrifugal turbines that operate at high turbine inlet temperatures are subjected to high thermal stresses. Thus, to minimize the bending stresses at the rotor inlet, the values of the flow coefficient, and loading coefficients are kept smaller than those already proposed [9,15]. However, for Organic Rankine Cycles (ORC), thermal stresses are substantially smaller due to the lower turbine inlet temperature values. Hence, the turbine's blade failure under the bending stresses will pose no threat while setting flow and loading coefficients on the higher side to improve the cycle's efficiency. Consequently, turbines operating in ORC could be optimized with high efficiency using higher values of flow and loading coefficients. In this context, Zheng et al. [15] conducted a design and optimization study based on meanline design calculations. Kim and Kim [16] performed a preliminary design study for a radial inflow turbine based on one-dimensional calculations. They have adopted new techniques of loading coefficients to achieve a design with higher efficiency. They have conducted 3D CFD simulations to predict the off-design performance of the turbine. However, the optimization procedure for the turbine design parameters is missing in the literature for small scale ORC turbines. To fill the gap, the current study was conducted to design and optimize the turbine for the same conditions and working fluid (R152a), as used by Kim and Kim [16]. For the design and optimization process, a MATLAB code was developed which was linked with the NIST REFPROP [17] to access the properties of the working fluid at various states. For the optimization process, a response surface methodology (RSM) procedure was adopted that utilized a genetic algorithm. Design variables considered for the current optimization study were rotational speed ($Ns$), total to static speed ratio ($v$), hub radius ratio $\left(\frac{r_{2h}}{r_3}\right)$, shroud radius ratio $\left(\frac{r_{2s}}{r_3}\right)$, inlet flow angle ($\alpha_2$) and number of blades ($n_b$). Various designs arranged by central composite design (CCD) were computed using TDC and verified by 3D-Reynolds-Averaged Navier-Stokes (RANS) simulations. Finally, the optimized design was tested, and the corresponding off-design turbine behavior was computed using 3D-RANS simulations through ANSYS-CFX [18].

## 2. Turbine Rotor Design Process

### 2.1. Turbine Design Guidelines

For a good design and optimization process, a preliminary good base-line design speeds up the process to achieve the final goal. In the current work, the following methodology was adopted to achieve the base design in order to initiate the optimization process.

#### 2.1.1. Design of the Rotor Inlet

The current section explains the procedure adopted to calculate the velocity triangle at the inlet of the turbine rotor and other associated parameters. It is concluded from the design calculations that the rotor inlet angle, that matches the incoming flow from inlet guide vane does not result in maximum efficiency of the turbine. However, it corresponds to negative incidence angles that may range from $-20°$ to $-30°$ [19,20].

For the velocity triangle with the flow entering the rotor at negative flow angle $\beta_2$, the negative incident angle is shown in Figure 1. Considering design point conditions when the flow exits the turbine axially, the Euler equation for the turbine is reduced to:

$$\frac{\dot{W}}{\dot{m}} = U_2 C_{\theta 2} \tag{1}$$

Referring to design point conditions $S_w = \frac{\dot{W}}{\dot{m} \, h_{10}}$, $h_{10} = \frac{a_{10}^2}{\gamma - 1}$, so (1) becomes

$$\left(\frac{U_2}{a_{1o}}\right)\left(\frac{C_{\theta 2}}{a_{1o}}\right) = \left(\frac{S_w}{\gamma - 1}\right) \tag{2}$$

It can be perceived from the velocity triangle that

$$C_{\theta 2} = C_{m2} \tan \beta_2 + U_2$$

$$C_{\theta 2} = \frac{C_{\theta 2}}{\tan \alpha_2} \tan \beta_2 + U_2$$

Inserting the value of $C_{\theta 2} = C_2 \, Sin \, \alpha_2$ and then rearranging the equation above gives

$$\tan^2 \alpha_2 \left(1 - \frac{U_2 C_{\theta 2}}{a_{o,1}^2} \frac{a_{o,1}^2}{C_2^2}\right) - \tan \alpha_2 \tan \beta_2 - \frac{U_2 C_{\theta 2}}{a_{o,1}^2} \frac{a_{o,1}^2}{C_2^2} = 0 \tag{3}$$

The above equation can be solved for the minimum value of the Mach number $M_{2min}$. It should be noted here that $M_2 = C_2/a_o$, where $a_o$ is velocity of sound at stagnation conditions. Further details on the derivation can be found in [21–23].

$$M_{2min} = \left(\frac{S_w}{\gamma - 1}\right)\left(\frac{2 \cos \beta_2}{(1 + \cos \beta_2)}\right) \tag{4}$$

To better understand the rotor inlet Mach number ($M_{2min}$), the above equation is plotted in Figure 2 as function of power ratio ($S_w$). It displays the minimum value of the inlet Mach number corresponding to any value of $S_w$ for various values of $\beta_2$ that vary from $-20°$ to $-60°$. Now, it is at the will of the designer to stipulate any inlet Mach numbers above the minimum that, however, would result in the increased value of the stator losses, due to higher velocities in the rotor that, in turn, increase the incidence losses from the design point, and this must be borne in mind. Furthermore, for the same value of $M_2$, the power ratio shows an increase in the negative $\beta_2$. In addition, it can be seen from

Figure 3 that value of the absolute flow angle decreases with the increase in the value of the inlet Mach number beyond its minimum permissible limit.

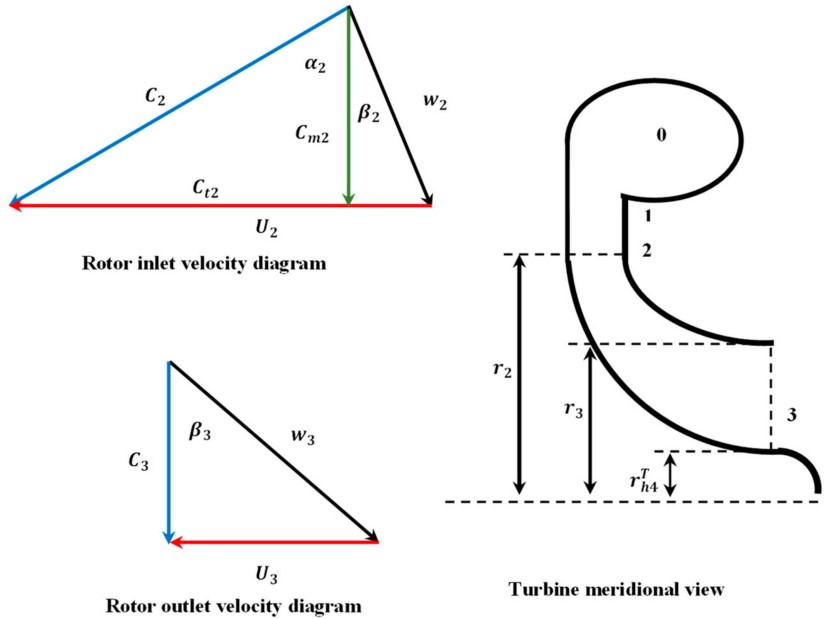

**Figure 1.** Velocity triangles with negative incident angle.

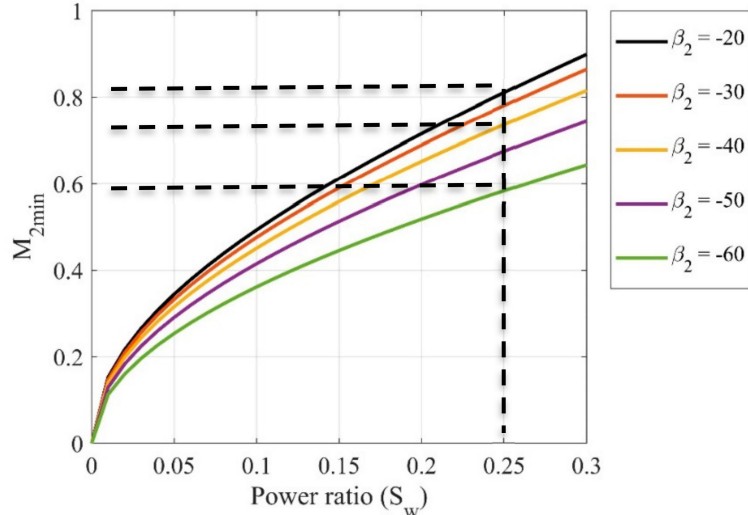

**Figure 2.** Variation of Mach number with power ratio for different values of $\beta_2$.

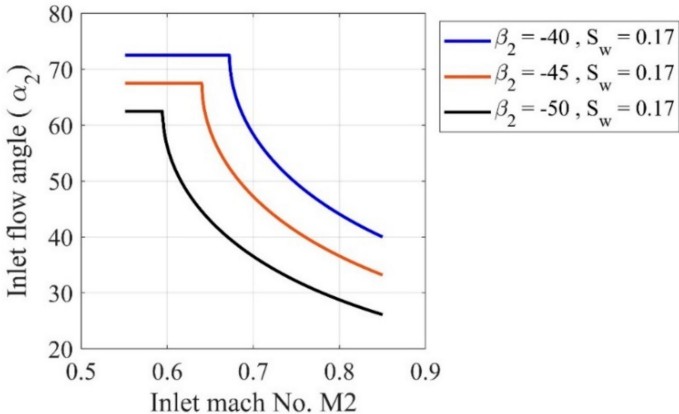

**Figure 3.** Effects of the Mach number on the inlet flow angle.

### 2.1.2. Design of Rotor Outlet

Considering zero discharge swirl at the outlet, the relative velocity at the shroud can be given by the following equation:

$$w_{3s}^2 = C_3^2 + U_{3s}^2 = C_3^2 + \omega^2\, r_{3s}^2 \tag{5}$$

Further details on the derivation can be found in [21–23]:

$$M_{3s}^2 = M_3^2 + \frac{\theta M_u^2}{1 - v^2} \frac{1}{M_3^2}\left(1 + \frac{\gamma - 1}{2}M_3^2\right)^{\frac{1}{2}}\left(\frac{T_{01}}{T_{03}}\right)^{\frac{1}{2}}\frac{p_{o1}}{p_{o3}} \tag{6}$$

Definitions of terms used in the above equation are as follows while further details can be found in [21–23]:

$$M_{3s}^{\cdot} = \frac{w_{3s}}{\sqrt{\gamma R T_3}} \tag{7}$$

$$M_3^{\cdot} = \frac{w_3}{\sqrt{\gamma R T_3}} \tag{8}$$

$$v = \frac{r_{3h}}{r_{3s}} \tag{9}$$

$$\frac{p_{o1}}{p_{o3}} = \left(1 - \frac{S_w}{\eta_{ts}}\right)^{-\frac{\gamma}{\gamma-1}} \tag{10}$$

$$M_u = \frac{U_2}{a_{01}} \tag{11}$$

Now Equation (6) can be written as below:

$$M_{3s}'^2 = M_3^2 + \frac{\theta M_u^2}{1 - v^2}\frac{1}{M_3^2}\left(1 + \frac{\gamma - 1}{2}M_3^2\right)^{\frac{1}{2}}\sqrt{\left(1 - \frac{S_w}{\eta}\right)}\left(1 - \frac{S_w}{\eta_{ts}}\right)^{-\frac{\gamma}{\gamma-1}} \tag{12}$$

The above equation is plotted in Figure 4 to show that the minimum value of relative Mach number exits against $\beta_3 = 55^0$ as indicated in the figure. At the same time, the minimum value of the absolute Mach number corresponds to a higher value of $\beta_3 = 70^0$. Thus, by selecting lower values of $\beta_3$, passage losses can be reduced while higher values of $\beta_3$ will help to minimize the exit losses. Therefore, a good compromise is always required between the values of absolute Mach number and relative Mach number as both are responsible for the certain losses in the turbine, i.e., exit and internal passage losses.

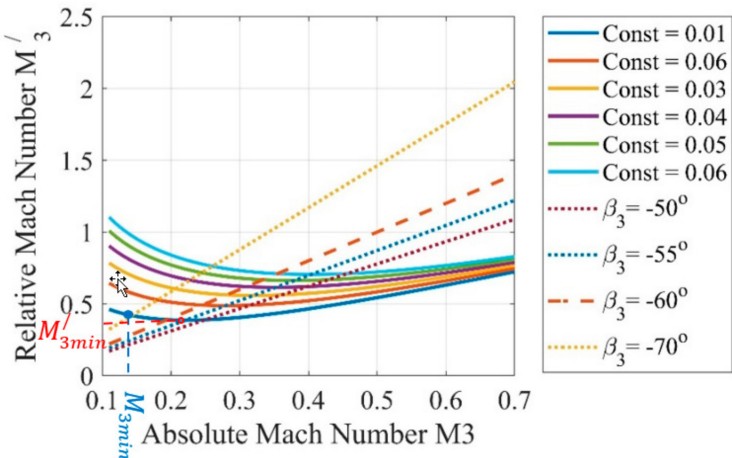

**Figure 4.** Variation in the value of Relative Mach number with absolute Mach number at various values of the exit blade angle.

*2.2. Meanline Design of the Turbine*

Meanline design calculations for the initial design were conducted using the methodology available in [22] that is based on the above-mentioned guidelines (Section 2.1). Rotational speed ($\omega$) was computed using Equation (13) using known value specific speed ($N_s$). $Q_3$ was estimated using the value of $\rho_3$ based on exit condition with the help of REFPROP using a known value of mass flow rate. The value of $\Delta h_s$ was taken from Equation (16).

$$N_s = \frac{\omega \sqrt{Q_3}}{(\Delta h_{id})^{0.75}} \tag{13}$$

The values of the discharge spout velocity were computed using Equation (14) that was used in Equation (15) along with the values of $v_{ts}$ to compute the value of rotor tangential speed $U_3$, while Equation (16) was used to compute the value of the $h_{04}$ using values of $\eta_{ts}$.

$$C_{0s} = \sqrt{2\Delta h_s} \tag{14}$$

$$U_3 = v_{ts}C_{0s} \tag{15}$$

$$h_{04} = h_{01} - \Delta h_s \eta_{ts} \tag{16}$$

The value of the rotor radius at the rotor inlet was computed using Equation (17) and rotor inlet stagnation pressure using Equation (18). Meanwhile, the value of the tangential component of the velocity was computed using Equation (19). Rotor thickness values at the inlet and exit were computed using Equations (20) and (21). Once the solution to Equation (19) is achieved, this is sufficient information to compute the inlet velocity triangle. Using the assumed value of $\alpha_2$, now $C_2, C_{r2}, W_2$ and $\beta_2$ could be computed. The values of the shroud and hub rotor radius at the exit were computed based on the assumptions given in Table 1. The value of the passage width ($b_2$) at the rotor inlet was computed using Equation (22). For that, the value of $\rho_2$ was computed as a function of $h_2$ and $s_2$ using REFPROP. The value of $s_2$ was computed as $s_{02} = s_2$ where $s_{02}$ was computed using REFPROP as a function of $P_{02}$ and $h_{02}$. Meanwhile, $h_2$ was computed using Equation (23).

$$r_2 = U_2/\omega \tag{17}$$

$$P_{02} = P_{01} - \frac{\rho_{01}\Delta h_{0,idl}(1 - \eta_s)}{4} \tag{18}$$

$$C_{\theta 2} = \frac{U_2 \, \eta_s}{2v_s^2} \tag{19}$$

$$t_{b2} = 0.04 \, r_2 \tag{20}$$

$$t_{b3} = 0.02 \, r_2 \tag{21}$$

$$b_2 = \frac{\dot{m}}{2 \, \pi \, r_2 \rho_2 C_{r2}} \tag{22}$$

$$h_2 = h_{02} - \frac{1}{2} C_2^2 \tag{23}$$

**Table 1.** Design parameters assumed for the base design.

| Turbine Inlet Temperature | $T_{01}$ | 299$K$ |
|---|---|---|
| Turbine inlet total pressure | $p_{01}$ | 545.89 $kPa$ |
| Turbine outlet static pressure | $p_3$ | 372.71 $MPa$ |
| Mass flow rate | $\dot{m}$ | 20 $kgs^{-1}$ |
| Isentropic specific power | $\Delta h_s$ | 11.18 $kJ \, Kg^1$ |
| Specific speed | $N_s$ | 0.45 |
| Total to static velocity ratio | $v_{ts}$ | 0.7 |
| Total to static efficiency first iteration | $\eta_{ts}$ | 0.8 |
| Shrouds exit to rotor inlet radius | $\frac{r_{s3}}{r_2}$ | 0.65 |
| Hub exit to rotor inlet radius | $\frac{r_{h3}}{r_2}$ | 0.2 |
| Number of rotor blades | $Z_R$ | 19 |
| Rotor inlet absolute velocity angle | $\alpha_2$ | $70^0$ |

The value of $C_{a3}$ was computed using Equation (24) using an assumed value of $\rho_3$. Meanwhile, the definitions of $r_3$ and $b_4$ are given in Equations (14) and (15). Once the value of $C_{a3}$ is known, Equation (27) was used to determine the value of $h_3$. The value of $\rho_3$ now can be corrected using REFPROP as a function of $\rho_3$ and $p_3$ that was converged through an iterative procedure between Equations (24)–(27).

$$C_{a3} = \frac{\dot{m}}{2\pi r_3 b_3 \rho_3} \tag{24}$$

$$r_3 = \frac{(r_{h3} + r_{s3})}{2} \tag{25}$$

$$b_3 = (r_{s3} - r_{h3}) \tag{26}$$

$$h_3 = h_{03} - \frac{1}{2} C_{a3}^2 \tag{27}$$

With assumed values of density at exit $\rho_4$, $C_{m4}$ was computed using Equation (32) based on definition and $b_4 = (r_{s5} - r_{h5})$. Furthermore, $h_4$ was computed using Equation (34) with known values of $h_{04}$ from Equation (33) and $\rho_4$ was corrected using REFPROP as a function of $h_4$ and $p_4$ through an iterative process. Once $\rho_4$ was corrected, all other state variables were computed using REFPROP and a function of $\rho_4$ and $p_4$. At this stage, with known information, the exit velocity triangle and rotor sizing were complete with the assumed value of efficiency.

### 2.2.1. Rotor Loss Model

The rotor loss model [23] presented in the current section was used to compute the different types of enthalpy losses occurring while the flow expands in the turbine rotor. The rotor loss model presented in the current section was used to compute the updated value of efficiency based on Equations (28) and (29), whereas efficiency used in the first iteration had an assumed value. Thus, this procedure was

recalculated between the rotor sizing model and rotor loss model until efficiency values converged. The flow diagram of the process is shown in Figure 2.

$$\Delta h_{loss} = \Delta h_{friction} + \Delta h_{secondary} + \Delta h_{tip\ clearance} + \Delta h_{exit} \tag{28}$$

$$\eta_{ts}^{corrected} = \frac{\Delta h_{loss}}{\Delta h_{loss} + \Delta h_{actual}} \tag{29}$$

### 2.2.2. Friction Losses

The term $\Delta h_{friction}$ in Equation (28) is associated with the losses associated with the viscous flow in the rotor and was computed using Equation (19). The values of $Re$, $l_{hyd}$, $d_{hyd}$ and $f$ are given by Equations (31)–(35).

$$\Delta h_{friction} = f\left(1 + 0.075\ Re^{0.25}\sqrt{\frac{d_{hyd}}{2r_c}}\right)\left[Re\left(\frac{d_2}{2r_c}\right)^{0.05}\right]\left[\frac{\left(W_2 + \frac{W_{s3}+W_{h3}}{2}\right)}{2}\right]\frac{l_{hyd}}{d_{hyd}} \tag{30}$$

$$Re_{nozzle} = \frac{\frac{U_2 b_2 \rho_2}{\mu_2} + \frac{U_3 b_3 \rho_3}{\mu_3}}{2} \tag{31}$$

$$l_{hyd} = \frac{\pi}{4}\left[\left(z - \frac{b_3}{2}\right) + \left(r_3 - r_{s4} - \frac{b_4}{2}\right)\right] \tag{32}$$

$$d_{hyd} = \frac{1}{2}\left[\left(\frac{4\pi r_4 b_3}{2\pi r_3 + n_b b_3}\right) + \left(\frac{2\pi\left(r_{s4}^2 - r_{h4}^2\right)}{\pi(r_{s5} - r_{h5}) + n_b b_4}\right)\right] \tag{33}$$

$$f = 8\left[\left(\frac{8}{Re_{nozzle}}\right)^{12} + \left(\left[2.457\ln\left(\frac{1}{\left(\frac{7}{Re_{nozzle}}\right)^{0.9} + 0.27\ RR}\right)\right]^{16} + \left[\frac{37530}{Re_{nozzle}}\right]^{16}\right)^{-1.5}\right]^{\frac{1}{12}} \tag{34}$$

$$c = \frac{z}{\cos\overline{\beta}} \text{ where } \tan\overline{\beta} = \frac{1}{2}(\tan\beta_3 + \tan\beta_4) \tag{35}$$

### 2.2.3. Tip Clearance Losses

Enthalpy losses due to flow leakage from the gap between the rotor tip and shroud casing are referred to as tip clearance losses. Tip clearance losses were computed using Equation (36)

$$\Delta h_{tip\ clearance} = \frac{U_3^3 n_b}{8\pi}\left(0.4\ \varepsilon_x C_x + 0.75\varepsilon_r C_r - 0.3\sqrt{\varepsilon_x \varepsilon_r C_x C_r}\right) \tag{36}$$

where

$$C_x = \frac{1 - \left(\frac{r_{s4}}{r_3}\right)}{C_{m3}b_3} \tag{37}$$

$$C_r = \left(\frac{r_{s4}}{r_3}\right)\frac{z - b_4}{C_{m6}r_4 b_4} \tag{38}$$

$$\varepsilon_x = \varepsilon_r = 0.02(r_{s4} - r_{h4}) \tag{39}$$

### 2.2.4. Exit and Secondary Losses

Exit losses are defined by the enthalpy loss associated with the exit velocity. The greater the exit velocity, the greater the exit losses. It is always intended to reduce the exit Mach number to reduce the

exit losses. Exit losses were computed by Equation (40), whereas secondary losses were computed by Equation (41).

$$\Delta h_{exit} = \frac{1}{2} \, C_3^2 \tag{40}$$

$$\Delta h_{secondary} = \frac{C_4 d_4}{Z_{rotor} r_c} \tag{41}$$

## 3. Response Surface Optimization

Response surface methodology (RSM) [24] is capable of the optimization processes when the response is affected by several design variables due to its accuracy, robustness, smaller numerical cost, and simplicity in implantation [25]. A number of studies are presented in the literature where response surface methodology (RSM) is chosen for the optimization process when numerous design variables are engaged [25]. Madsen et al. [26] employed RSM for the optimization of a diffuser. On the other hand, Ahn et al. [27] make use of it for the optimization of an airfoil design. Built on the above debate RSM was implemented for the present analysis to improve the rotor geometry for the parameters given in Table 2.

**Table 2.** List of the design variables with the upper and lower limits.

| Input Design Variable for the Optimization Procedure | | Lower Limit | Upper Limit |
|---|---|---|---|
| Speed ratio ($v_{ts}$) | $x_1$ | 0.65 | 0.8 |
| Absolute flow angle at rotor inlet ($\alpha_2$) [degree] | $x_2$ | 50 | 80 |
| Rotational speed | $x_3$ | 2000 | 5000 |
| $\left(\frac{r_{h3}}{r_2}\right)$ | $x_4$ | 0.55 | 0.80 |
| $\left(\frac{r_{s3}}{r_2}\right)$ | $x_5$ | 0.15 | 0.30 |

The relation between the response variable (power) and design factors was chosen using a second-degree model given by Equation (26)

$$f(x) = \alpha_0 + \sum_{i=1}^{n} \beta_{i,i} x_i^2 + \sum_{i=1}^{n-1} \sum_{j=i+1}^{nd} \gamma_{i,j} x_i x_j + \sum_{i=1}^{n} \gamma_i x_i \tag{42}$$

In the above equation $x_i$ and $f(x)$ correspond to design variables and responses, respectively, whereas $\alpha_0, \beta_{i,i}, \gamma_{i,j}$ and $\gamma_i$ are regression coefficients. Moreover, term $n$ corresponds to the total number of design variables (e.g., for this study; $x_1 \to x_5$) that are listed in the Table 2.

For the design of experiments, the central composite design (CCD) option was used. This option enables the calculation of the overall trends of the Meta-model based on a screening set to better direct the choice of options in Optimal Space-Filling Design. This option could handle up to 20 different design variables. The central composite design (CCD) contains an embedded factorial or fractional factorial design with center points that is augmented with a group of 'star points' that allow estimation of curvature [24]. If the distance from the center of the design space to a factorial point is ±1 unit for each factor, the distance from the center of the design space to a star point is $|\alpha| > 1$. The precise value of $\alpha$ depends on certain properties desired for the design and on the number of factors involved. To maintain rotation ability, the value of $\alpha$ depends on the number of experimental runs in the factorial portion of the central composite design:

$$\alpha = (2^n)^{\frac{1}{4}}$$

Further details on the response surface methodology and central composite design can be found in the literature [24]. The design variable considered in the current study with defined upper and lower limits is listed in Table 2. The design of experiment calculated using central composite design (CCD) is listed in Table 3. It should be noted here that each combination of the design variables listed in Table 3 represents a distinct design of the turbine. The design of the turbine for each set of parameters listed in

Table 3 was computed using the meanline turbine design code (Figure 5). Now computed power of the turbine as a response variable f(x) for all designs listed in Table 3 can be used to solve the Equation (26) through regression analysis.

**Table 3.** Design of experiment using Central Composite Design (CCD).

| S.No. | $\alpha_2$ | $\frac{r_{s3}}{r_2}$ | $\frac{r_{h3}}{r_2}$ | *RPM* | $\nu_{ts}$ |
|-------|-------|------|------|------|------|
| 1 | 65.00 | 0.65 | 0.22 | 3500 | 0.68 |
| 2 | 50.00 | 0.65 | 0.22 | 3500 | 0.68 |
| 3 | 57.50 | 0.65 | 0.22 | 3500 | 0.68 |
| 4 | 80.00 | 0.65 | 0.22 | 3500 | 0.68 |
| 5 | 72.50 | 0.65 | 0.22 | 3500 | 0.68 |
| 6 | 65.00 | 0.55 | 0.22 | 3500 | 0.68 |
| 7 | 65.00 | 0.60 | 0.22 | 3500 | 0.68 |
| 8 | 65.00 | 0.75 | 0.22 | 3500 | 0.68 |
| 9 | 65.00 | 0.70 | 0.22 | 3500 | 0.68 |
| 10 | 65.00 | 0.65 | 0.15 | 3500 | 0.68 |
| 11 | 65.00 | 0.65 | 0.19 | 3500 | 0.68 |
| 12 | 65.00 | 0.65 | 0.30 | 3500 | 0.68 |
| 13 | 65.00 | 0.65 | 0.26 | 3500 | 0.68 |
| 14 | 65.00 | 0.65 | 0.22 | 2000 | 0.68 |
| 15 | 65.00 | 0.65 | 0.22 | 2750 | 0.68 |
| 16 | 65.00 | 0.65 | 0.22 | 5000 | 0.68 |
| 17 | 65.00 | 0.65 | 0.22 | 4250 | 0.68 |
| 18 | 65.00 | 0.65 | 0.22 | 3500 | 0.55 |
| 19 | 65.00 | 0.65 | 0.22 | 3500 | 0.61 |
| 20 | 65.00 | 0.65 | 0.22 | 3500 | 0.80 |
| 21 | 65.00 | 0.65 | 0.22 | 3500 | 0.74 |
| 22 | 50.00 | 0.55 | 0.15 | 2000 | 0.80 |
| 23 | 57.50 | 0.60 | 0.19 | 2750 | 0.74 |
| 24 | 80.00 | 0.55 | 0.15 | 2000 | 0.55 |
| 25 | 72.50 | 0.60 | 0.19 | 2750 | 0.61 |
| 26 | 50.00 | 0.75 | 0.15 | 2000 | 0.55 |
| 27 | 57.50 | 0.70 | 0.19 | 2750 | 0.61 |
| 28 | 80.00 | 0.75 | 0.15 | 2000 | 0.80 |
| 29 | 72.50 | 0.70 | 0.19 | 2750 | 0.74 |
| 30 | 50.00 | 0.55 | 0.30 | 2000 | 0.55 |
| 31 | 57.50 | 0.60 | 0.26 | 2750 | 0.61 |
| 32 | 80.00 | 0.55 | 0.30 | 2000 | 0.80 |
| 33 | 72.50 | 0.60 | 0.26 | 2750 | 0.74 |
| 34 | 50.00 | 0.75 | 0.30 | 2000 | 0.80 |
| 35 | 57.50 | 0.70 | 0.26 | 2750 | 0.74 |
| 36 | 80.00 | 0.75 | 0.30 | 2000 | 0.55 |
| 37 | 72.50 | 0.70 | 0.26 | 2750 | 0.61 |
| 38 | 50.00 | 0.55 | 0.15 | 5000 | 0.55 |
| 39 | 57.50 | 0.60 | 0.19 | 4250 | 0.61 |
| 40 | 80.00 | 0.55 | 0.15 | 5000 | 0.80 |
| 41 | 72.50 | 0.60 | 0.19 | 4250 | 0.74 |
| 42 | 50.00 | 0.75 | 0.15 | 5000 | 0.80 |
| 43 | 57.50 | 0.70 | 0.19 | 4250 | 0.74 |
| 44 | 80.00 | 0.75 | 0.15 | 5000 | 0.55 |
| 45 | 72.50 | 0.70 | 0.19 | 4250 | 0.61 |
| 46 | 50.00 | 0.55 | 0.30 | 5000 | 0.80 |
| 47 | 57.50 | 0.60 | 0.26 | 4250 | 0.74 |
| 48 | 80.00 | 0.55 | 0.30 | 5000 | 0.55 |
| 49 | 72.50 | 0.60 | 0.26 | 4250 | 0.61 |
| 50 | 50.00 | 0.75 | 0.30 | 5000 | 0.55 |

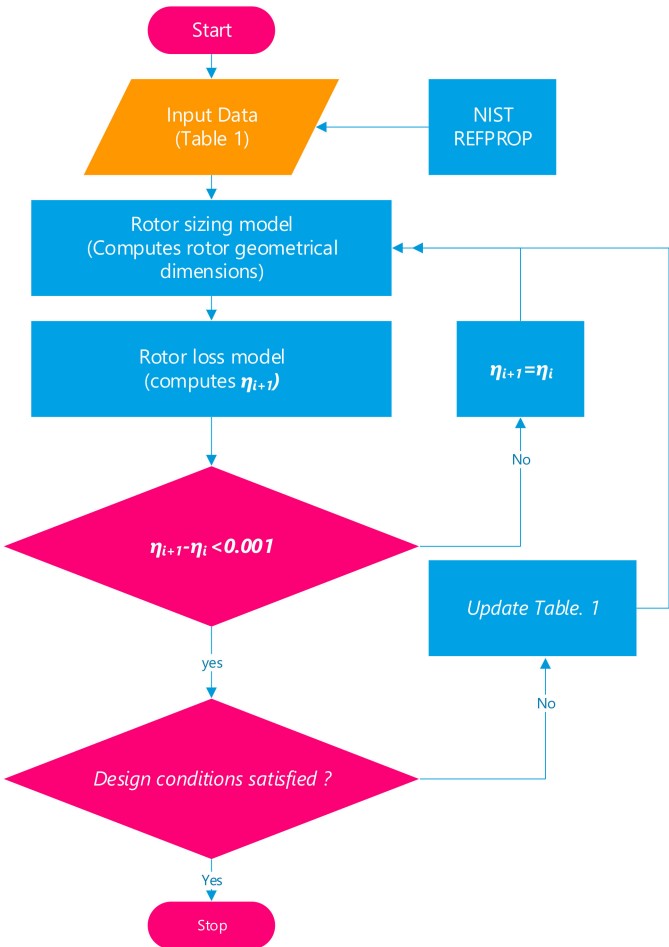

**Figure 5.** Flow diagram for the meanline design process.

## 4. Computation Model

### 4.1. Turbine Geometry/Meshing

Optimized geometry computed through meanline design calculations and response surface optimization was constructed using the ANSYS-blade editor. Figure 6 shows the 3D model of the turbine geometry along with the meridional view and distribution of blade angles and blade thickness with streamline direction.

Once the 3D model of optimized geometry was constructed, it was exported to the ANSYS Turbine grid for mesh generation, where its high-quality hexahedral mesh was generated. The number of nodes in the inlet and outlet regions was fixed to 40 for each. Along with the streamwise direction, the number of nodes was set as 150 while along the blade height the number of nodes was set to 80. In the tip clearance, the region number of nodes was fixed as 15 while in the radial direction for one passage the number of nodes was fixed to 125. The distribution of nodes explained above was reached through a mesh optimization study. According to the study five different meshes were constructed and the mesh finally chosen showed no further variation of results by changing the number of nodes. The number of the nodes of the optimized mesh was 1,555,263 nodes. A 3D mesh of the computational domain is shown in Figure 7.

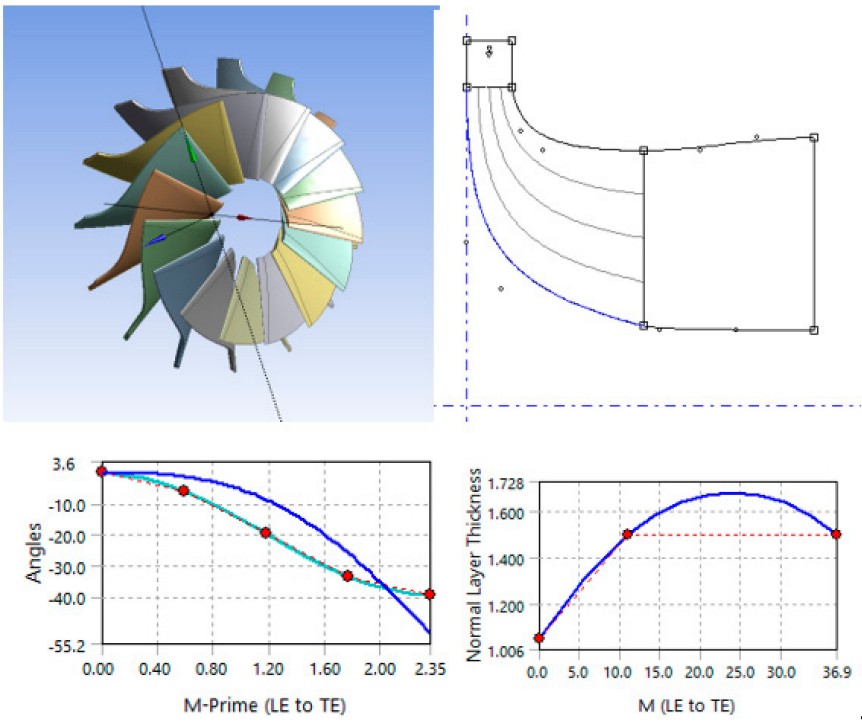

**Figure 6.** 3D geometry of the optimized model, meridional view and its angle thick distributions.

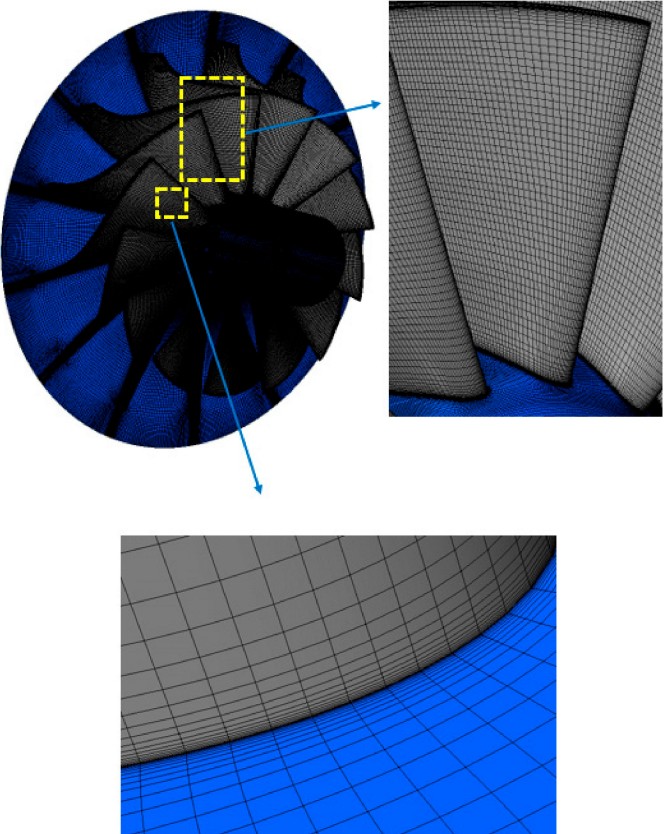

**Figure 7.** 3D mesh of the computational domain.

### 4.2. Computation Method and Boundary Equations

To solve the computational domain shown in Figure 6, a commercial code ANSYS–CFX was employed. The steady Reynolds averaged Navier–Stokes equation was solved to compute the flow and energy equation to solve the temperature field. To resolve the Reynolds stress term, the shear stress transport (SST) turbulence model was used. Only one passage was solved to reduce the computational cost. Total pressure inlet and static pressure outlet conditions [28] were specified while the blade, hub, and shrouds were assigned as a wall. The blade and hub were rotating walls, but the shroud was assigned as a stationary wall. Rotational periodic conditions were set to the downstream and upstream surfaces.

## 5. Results

### 5.1. Rotor Optimization

As discussed in the above section, the five design variables, i.e., rotational speed ($Ns$), total to static speed ratio ($v$), hub radius ratio ($\frac{r_{2h}}{r_3}$), shroud radius ratio ($\frac{r_{2s}}{r_3}$) and inlet flow angle ($\alpha_2$) were considered for the optimization process. The lower and upper limits of the design variables are given in Table 3. In addition, various designs computed using the central composite design are given in Table 2. Sensitivity results based on the soled regression model given by Equation (42), for the design variables, i.e., rotational speed ($Ns$), total to static speed ratio ($v$), hub radius ratio ($\frac{r_{2h}}{r_3}$), shroud radius ratio ($\frac{r_{2s}}{r_3}$) and inlet flow angle ($\alpha_2$); for the response variables, i.e., power output, load losses, clearance losses, friction losses, exit losses, and incidence losses, are given in Figure 8. The results suggest that response variables are most sensitive to inlet flow angle ($\alpha_2$) and speed ratio ($v$).

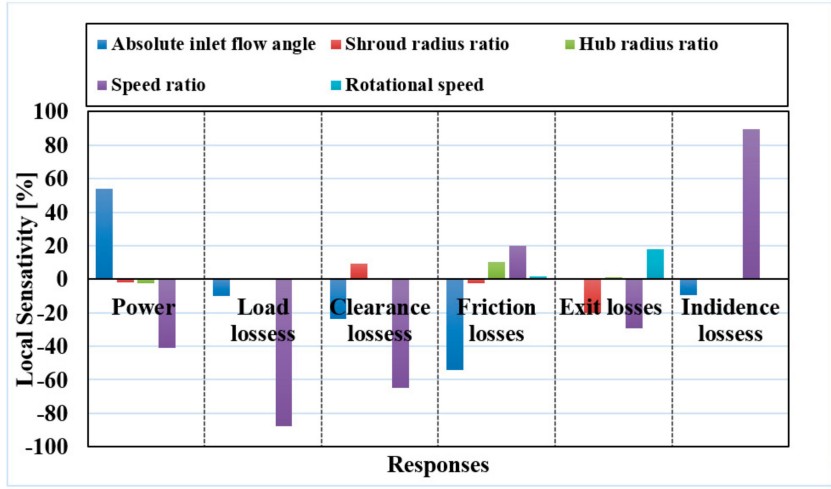

**Figure 8.** Sensitivity analysis of design variables.

5.1.1. Response Surfaces Bounded by $v_{ts}$ and Rotor Rotational Speed

The response surfaces for the optimization process computed by solving the RSM model (Section 3) are shown in Figures 9–15. Each of the Figures 9–15 show variation in the values of the two design variables to examine their effect on the response variable. It should be noted here that, while the response of the two variables was studied, other design variables were kept constant. Variation of the power with the total to static velocity ratio and rotor rotational speed is shown in Figure 9. It could be said that power increases with the rise in the value of $v_{ts}$ initially, and then it starts to decrease after reaching the peak. This trend can be seen for all the values of the rotational speed. It is clear for the response surface that power is more sensitive to the value of $v_{ts}$, lower values of the rotor RPM, while its sensitivity to $v_{ts}$ decreases as the rotational speed increases. On the other hand, power increases with the increase in the value of rotational speed attains its maximum value before declining with further increases in rotational speed. It is obvious from the response surface that power is more sensitive to

rotational speed at higher values of the total to static velocity ratio ($\nu_{ts}$). The region corresponding to the maximum power is shown in the figure by a red circle and arrow.

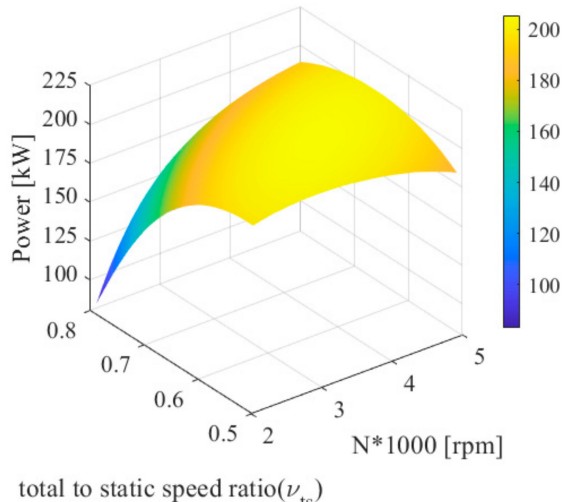

**Figure 9.** Variation of the power with the total to static speed ratio and rotor RPM.

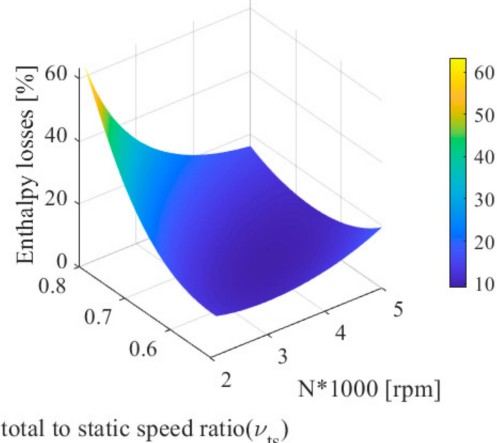

**Figure 10.** Variation of the enthalpy losses with the total to static speed ratio and rotor RPM.

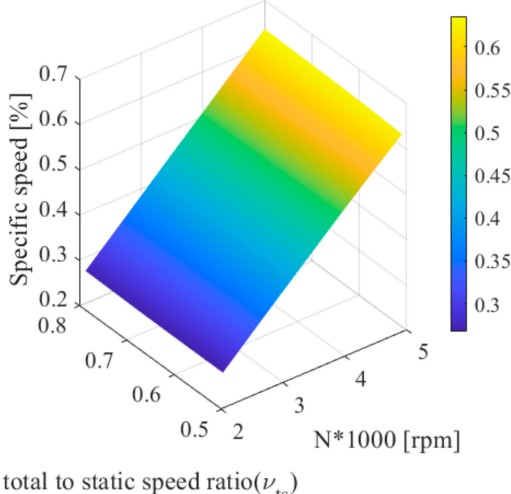

**Figure 11.** Variation of the specific speed with total to static speed ratio and rotor RPM.

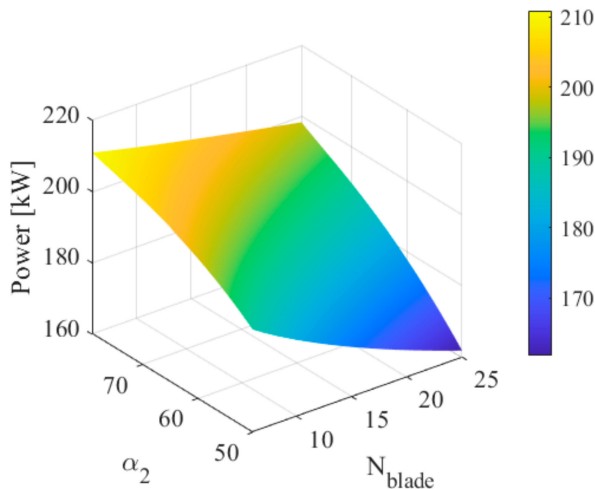

**Figure 12.** Variation of the power with $\alpha_2$ and number of blades.

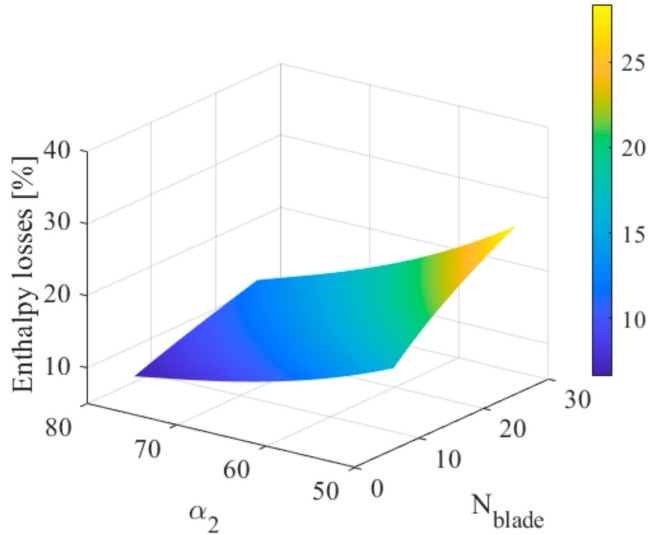

**Figure 13.** Variation of enthalpy losses with $\alpha_2$ and number of blades.

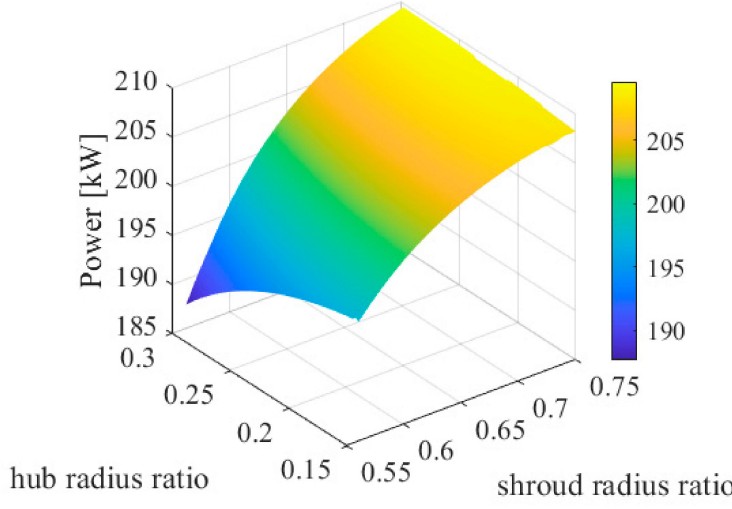

**Figure 14.** Variation of the power with shroud radius ratio and hub radius ratio.

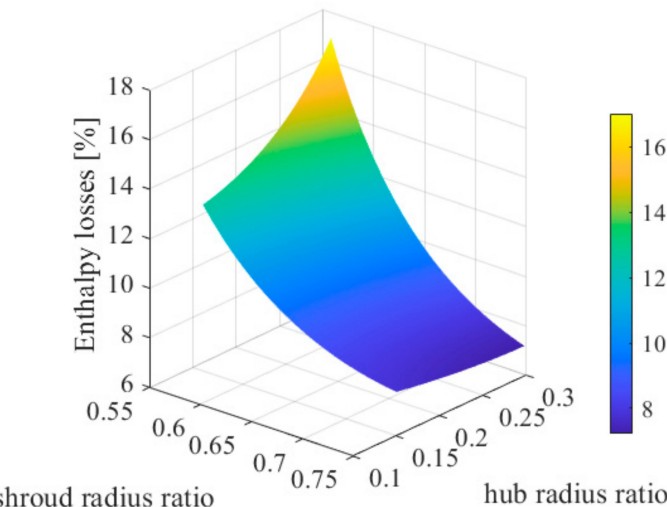

**Figure 15.** Variation of the enthalpy losses with shroud radius ratio and hub radius ratio.

Variation of enthalpy losses with $v_{ts}$ and rotational speed is shown in Figure 10. According to the figure, enthalpy losses decrease and then increase with the increase in the value of rotor's revolutions per minute (RPM) for a given value of $v_{ts}$. Similarly, losses decrease initially and then start increasing with the increase in the value of the $v_{ts}$ at a given fixed value of the $N$. Figure 11 shows the variation of the specific speed with rotational speed and $v_{ts}$. The specific speed of the rotor increases linearly with the increase in rotor rpm, significantly. The specific speed also increases with the increase in the value of $v_{ts}$, however, its sensitivity to $v_{ts}$ is smaller in comparison with its sensitivity to rotational speed.

### 5.1.2. Response Surfaces Bounded by $\alpha_2$ and Number of Blades

The response surface of power bounded by $\alpha_2$ and the number of blades is shown in Figure 12. It can be seen that power increases linearly with the increase in $\alpha_2$. The sensitivity of power at higher values of the number of blades is higher in comparison to its sensitivity corresponding to a fewer number of blades. On the other hand, power decreases linearly by increasing the number of blades at lower values of $\alpha_2$ but at higher values of $\alpha_2$ the power increases initially with a slight increase in the value of the number in blades and then starts to decrease. Concerning the value of the enthalpy loss increase with the increase in the number of blades at smaller values of $\alpha_2$, but at higher values of $\alpha_2$, losses decrease slightly initially and then start increasing with the increase in the number of blades, as shown in Figure 13. Hence, enthalpy losses initially reduce with the increase in the value of $\alpha_2$ and then start rising at a given value of the number of rotor blades.

### 5.1.3. Response Surfaces Bounded by Hub Radius Ratio and Shroud Radius Ratio

Figures 14 and 15 show the response surface bounded by $\frac{r_{h3}}{r_2}$ and $\frac{r_{s3}}{r_2}$. It can be seen in Figure 14 that the response surface of the power is more sensitive to the shroud radius ratio $\frac{r_{s3}}{r_2}$ in comparison with its sensitivity to $\frac{r_{h3}}{r_2}$. Power increases rapidly with the increase in the value of $\frac{r_{s3}}{r_2}$ initially, and later on its sensitivity to a further rise in $\frac{r_{s3}}{r_2}$ decreases significantly. The power response to the hub radius ratio exhibits an interesting trend: power increases with the increase in the value of $\frac{r_{h3}}{r_2}$ at higher values of $\frac{r_{s3}}{r_2}$, but shows an opposite trend at lower values of $\frac{r_{s3}}{r_2}$. Figure 15 shows a response surface of enthalpy losses bounded by $\frac{r_{h3}}{r_2}$ and $\frac{r_{s3}}{r_2}$. It could be depicted from the response surface, similar to the power response surface, which is more sensitive to $\frac{r_{s3}}{r_2}$ in comparison with $\frac{r_{h3}}{r_2}$.

5.1.4. Comparison of Meanline Design/Optimized Design with CFD Results

A comparison of the results for the base meanline design and the optimized design is listed in Table 4. This comparison suggests that an optimized set of parameters causes a significant improvement in the performance of the turbine rotor. For the optimized design, an improvement of 4.94% and 4.29% was observed for the power and total to static efficiency, respectively. Meanwhile, Table 5 shows the differences in the comparison between the meanline design calculations and CFD calculations.

**Table 4.** Comparison of results for the meanline design and optimized design.

| Turbine's Geometry and Performance Parameters | Values (Baseline Design) | Values (Optimized Design) | Difference [%] |
|---|---|---|---|
| $(\nu_{ts})$ | 0.8 | 0.69 | −13.75 |
| $(\alpha_2)$ [degree] | 65 | 82 | 26.15 |
| $\left(\frac{r_{h3}}{r_2}\right)$ | 0.18 | 0.25 | 38.89 |
| $\left(\frac{r_{s3}}{r_2}\right)$ | 0.65 | 0.79 | 21.54 |
| Rotational speed [RPM] | 5000 | 3800 | −24.00 |
| $(\Delta h_s)$ $\left[\text{kJ kg}^{-1}\right]$ | 11.184 | 11.184 | 0.00 |
| Power [kW] | 194.6 | 204.22 | 4.94 |
| Total to static efficiency [%] | 87.01 | 91.3 | 4.93 |

**Table 5.** Comparison of meanline and CFD results for the computed optimized design.

| Parameter | Meanline Design Calculation | CFD | Deviation [%] |
|---|---|---|---|
| Total to total pressure ratio $\left(\frac{P0_2}{P_3}\right)$ | 1.439 | 1.411 | 1.94 |
| Total to static pressure ratio $\left(\frac{P0_2}{P_3}\right)$ | 1.465 | 1.465 | 0 |
| Mass flow rate $\dot{m}$ $\left[\text{kg s}^{-1}\right]$ | 20 | 20.61 | 3.05 |
| Total to static efficiency $(\eta_{ts})$ | 91.3 | 93.4 | 2.30 |
| Power output $\left(\dot{W}\right)$ | 204.22 | 215.29 | 5.42 |

The maximum difference was found for the power values that are 5.42%, while the difference for the mass flow rate, efficiency, and total to total pressure ratio was found at 3.05%, 2.30%, and 1.94%, respectively. Figure 16 shows velocity vectors on the 50% span, while Figure 17 shows entropy and pressure distribution at the plane location at a 50% span. It could be seen clearly in these figures that one of the reasons for improved efficiency for the optimized design is the recirculation zone near the incidence point of the blade, which is causing high entropy losses that were rectified in the improved design. Qualitative results are shown in Figure 18. Figure 18a shows the velocity streamlines in the turbine passage, which could show that flow accelerates as it moves toward the turbine exit plane. Pressure contours on the turbine exit plane are shown in Figure 18b. Figure 18c shows the pressure distribution on the blade to blade view that is constructed at a constant span value, i.e., span = 50% of the blade height. Figure 18d shows Mach number contours on the same location. The flow Mach number increases towards the outlet of the radial turbine and reaches its maximum value, i.e., 1.8, and then it starts decreasing in the diffuse section of the turbine. Figure 18c,d exhibits the nozzle nature of the turbine where pressure decreases and the Mach number increases in the streamwise direction. Higher Mach at the exit of the blade section cause flow separations that increase the entropy losses in the wake of a blade, as shown in Figure 18e.

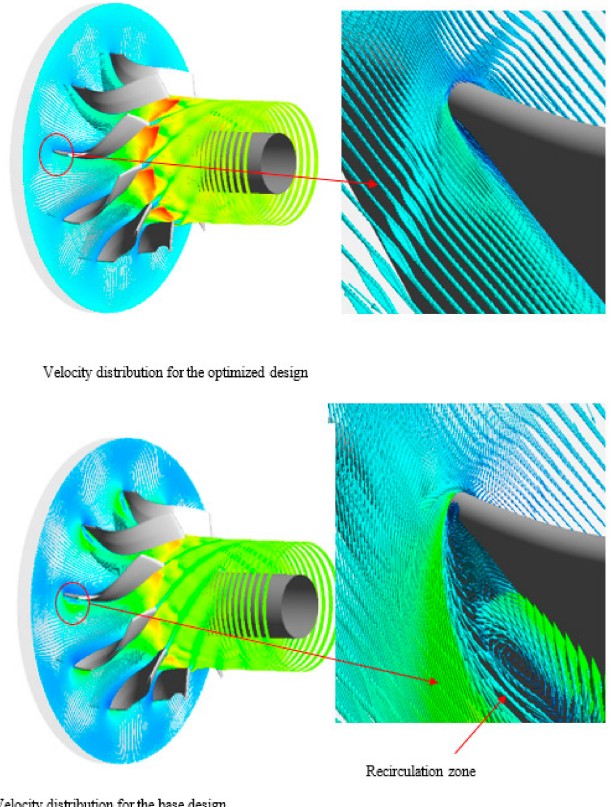

Figure 16. Velocity vectors on 50% span for the base and optimized design.

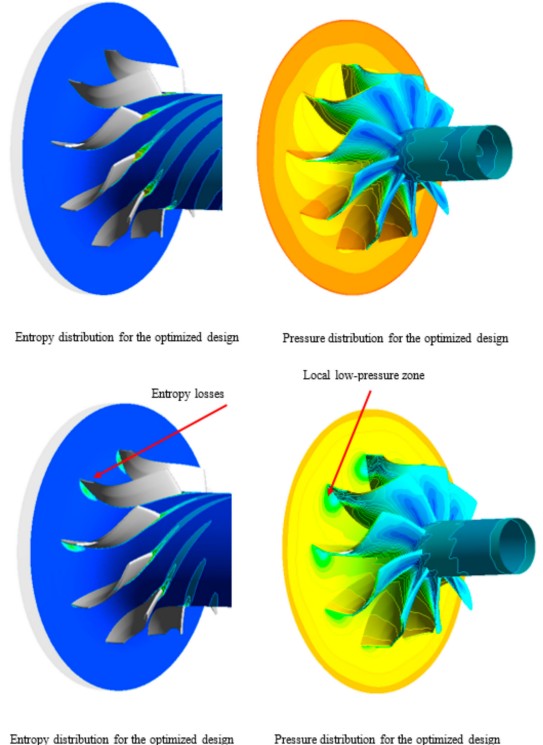

Figure 17. Entropy and pressure contours on 50% span for the base and optimized design.

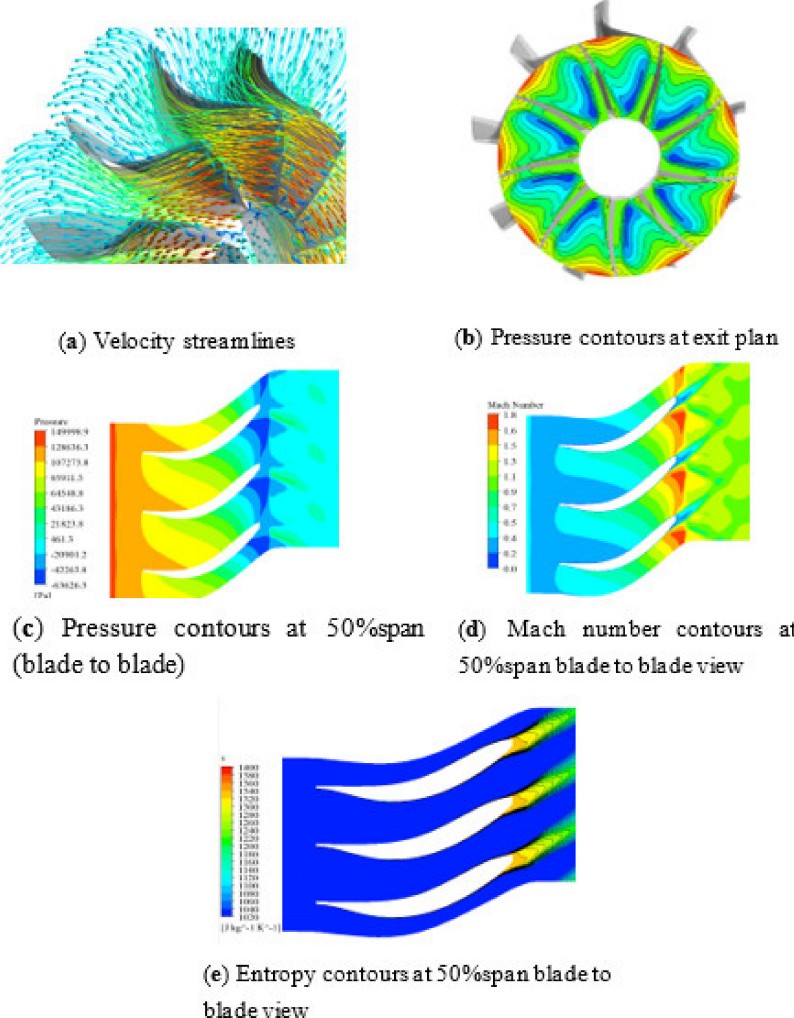

(a) Velocity streamlines

(b) Pressure contours at exit plan

(c) Pressure contours at 50%span (blade to blade)

(d) Mach number contours at 50%span blade to blade view

(e) Entropy contours at 50%span blade to blade view

**Figure 18.** (**a**) Velocity streamlines in the turbine passage, (**b**) pressure contours at the exit plane, (**c**) pressure contours at 50%span (blade to blade), (**d**) Mach number contours at 50%span blade to blade view, (**e**) Entropy contours at 50%span blade to blade view.

Figure 19 shows the off-design maps of the turbine. Off design maps were generated to ensure the behavior of the turbine during off-design conditions. Off-design maps showed that the behavior of the off-design conditions is appropriate and provided additional validity of the adopted process. This is based on the fact that the operational margin of the optimized turbine in the off-design conditions is sufficiently broad. It can be seen from Figure 19 that the optimized turbine can exhibit reasonable performance by altering the flow rate of the turbine by ±20% of its design point value.

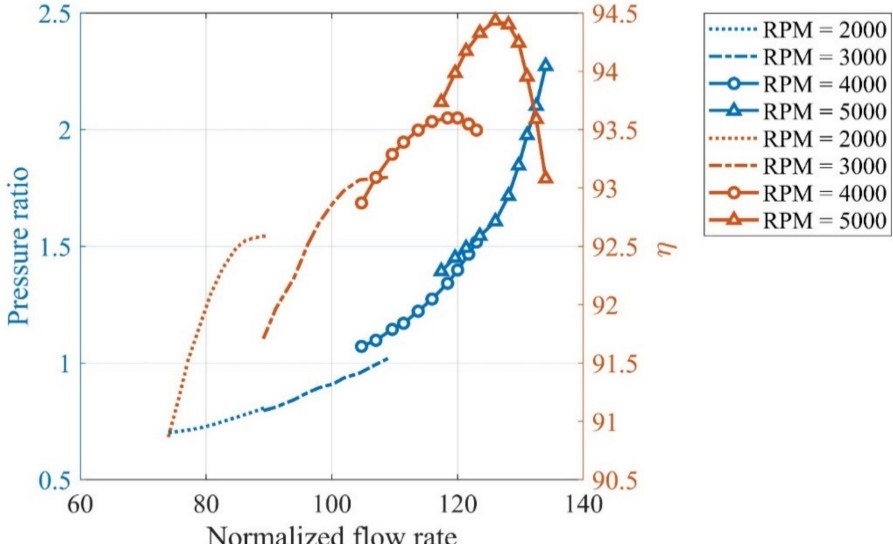

**Figure 19.** Off-design maps for the optimized turbine rotor.

## 6. Conclusions

The current study was conducted to design and optimize the turbine geometry for ocean thermal conversion systems by maximizing the efficiency of the turbine at low inlet temperature. Best practices for the design were discussed and implemented to improve and make the design and optimization process more robust. Meanline design was computed that was later optimized using the response surface optimization technique. The process was iterated between meanline methodology and 3D-RANS simulations until the required performance was achieved. The following deductions were made from the current study.

- For an accurate design calculation process, the accurate prediction of thermo—physical properties is crucial. For the current design process, MATLAB code was linked with NIST's REFPROP for the accurate provision of the properties. That is a distinct feature of the current study that has been ignored in similar studies found in the literature.
- It has been shown that a better meanline design based on the best design practices can significantly reduce the computation time and resources in the design and optimization of the turbine.
- The design optimization process is based on minimizing the losses, thus maximizing power and efficiency. For the current study, an improvement of 4.7% in power and 4.2% in efficiency is achieved through the optimization process.
- The maximum deviation in the meanline and CFD calculation was found for the output power of the turbine, which is 5.42%.
- Reported off-design results add to an additional validation toward the effectiveness of the procedure suggested for the design and estimation of a radial turbine.
- Response surface methodology (RSM) integrated with the 3D-RANS calculation is an effective, reliable, and robust method of the design and optimization of turbomachinery.

**Author Contributions:** Conceptualization, Methodology, Software Data curation, K.A.; Writing—riginal draft preparation, K.A., A.A. (Ali Ashour) and A.A. (Abdullah Alfalah); Visualization, Investigation, Y.A.; Software, Validation, Y.A. and A.A. (Abdullah Alfalah); Writing—Reviewing and Editing, K.A. and A.A. (Ali Ashourand). All authors have read and agreed to the published version of the manuscript.

**Funding:** This research received no external funding.

**Conflicts of Interest:** The authors declare that they have no known competing financial interests or personal relationships that could have appeared to influence the work reported in this paper.The authors declare the following financial interests/personal relationships which may be considered as potential competing interests.

## Nomenclature

| | |
|---|---|
| $a$ | speed of sound |
| $b$ | width of the blade [m] |
| $C$ | absolute speed $\left[\text{m s}^{-1}\right]$ |
| $C_p$ | specific heat capacity [J. $\text{kg}^{-1}.\text{K}^{-1}$] |
| $C_x$ | rotor tip clearance (axial) [m] |
| $C_r$ | rotor tip clearance (radial) [m] |
| $d_{hyd}$ | hydraulic diameter [m] |
| $f$ | friction factor |
| $h$ | specific enthalpy $\left[\text{J. kg}^{-1}\right]$ |
| $l_{hyd}$ | hydraulic length [m] |
| $\dot{m}$ | mass flow rate $\left[\text{kg s}^{-1}\right]$ |
| $M$ | Mach number |
| $N$ | rotational speed [rpm] |
| $N_s$ | specific speed |
| $p$ | pressure [Pa] |
| $Q$ | Flow rate $\left[\text{m}^3\text{s}^{-1}\right]$ |
| $r$ | Radius [m] |
| Re | Reynolds number |
| $S_w$ | power ratio |
| $t$ | blade thickness [m] |
| $T$ | temperature [K] |
| U | tangential speed of rotor $\left[\text{m s}^{-1}\right]$ |
| u | velocity $\left[\text{m s}^{-1}\right]$ |
| $w$ | relative speed $\left[\text{m s}^{-1}\right]$ |
| $\dot{W}$ | power [W] |
| $z$ | rotor length |

## Greek Symbols

| | |
|---|---|
| $\alpha$ | velocity angle (absolute) [degree] |
| $\beta$ | velocity angle (relative) [degree] |
| $\eta$ | efficiency $\left[\text{kg m}^{-1}\text{ s}^{-1}\right]$ |
| $\mu$ | dynamic viscosity $\left[\text{kg m}^{-1}\text{ s}^{-1}\right]$ |
| $\rho$ | density $\left[\text{kgm}^{-3}\right]$ |
| $\omega$ | angular speed $\left[\text{rad s}^{-1}\right]$ |
| $\gamma$ | specific heat capacity ratio |

## Sub and Super Scripts

| | |
|---|---|
| / | relative component |
| 0 | stagnation |
| 1 | nozzle inlet |
| 2 | rotor inlet |
| 3 | rotor exit |
| $b$ | blade |
| $c$ | cycle |
| $C$ | compressor |
| $h$ | hub |
| $max$ | maximum value |
| $min$ | minimum value |
| $s$ | Shroud, isentropic |
| $t$ | turbine |
| $\theta$ | tangential direction |

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
