# Peer review of "Design and Optimization of a Radial Turbine to Be Used in a Rankine Cycle Operating with an OTEC System"

_jmse, doi:10.3390/jmse8110855_

Round 1

Reviewer 1 Report

In this manuscript, a complete optimization process for an ORC radial turbine has been carried out. The base design, which is based on the meanline design methodology, is further optimized based on the CFD simulation results. The topic should be of interest to the jmse readers. The paper is well written, though there are some typo errors, for example

(1) In equation (28), there is an extra plus sign.

(2) Section 4.2, in the second line, “naïve stokes” should be “Navier Stoke”. In the same paragraph, what does the “stone wall” mean?

In summary, I would suggest accepting it for publication.

Author Response

Comments

Response

Added on

In this manuscript, a complete optimization process for an ORC radial turbine has been carried out. The base design, which is based on the meanline design methodology, is further optimized based on the CFD simulation results. The topic should be of interest to the jmse readers. The paper is well written, though there are some typo errors, for example

The authors are thankful for the detailed review and valuable observations/suggestions made by the reviewer that helped to improve the manuscript. All the concerns raised by the reviewer have been addressed in the following lines.

1.

In equation (28), there is an extra plus sign.

The equation has been updated and highlighted in the manuscript

Page 7- Eq. (28)

2.

Section 4.2, in the second line, “naïve stokes” should be “Navier Stoke”. In the same paragraph, what does the “stone wall” mean?

The correction has been made. Now the quoted text has been replaced with the Reynolds averaged Navier-Stokes equation and highlighted in the updated manuscript.

The “Stone wall” term is a typo that has been now replaced with a “stationary wall”.

Page 11

3.

In summary, I would suggest accepting it for publication.

Reviewer 2 Report

It is worth adding in abstract, at least, two sentences about the results of the research, what was obtained (not just a description of what was done).

Author Response

(The authors gave the same response as above.)

Reviewer 3 Report

The authors have conducted an interesting research work. They have described their motivation for their in the introduction and then the implementation a design optimization method for radial turbines. This is based on 1D meanline preliminary design, as well as in CFD and an optimization method. Evaluation of the optimized design is also made by means of 3D Navier-Stokes numerical simulations. The whole idea is good and the research topic modern and interesting. However, this reviewer judges that major revision is required in order the paper to be published. In particular, improvements are required that concern the adequate description of the method used and the clear presentation of results. Relevant remarks and details are given below.

  • The title of the paper is not adequate; it is rather misleading. It actually concerns ‘Design and Optimization of a Radial Turbine’. The second part of the title (‘… for a Rankine Cycle Operating with Ocean Thermal Conversion System’) is just the motivation for this work. Apart from the literature review in the introduction, no other reference is made in the paper to the issues mentioned in the second part of the title. There is no link of the design of the turbine with the Rankine cycle or justification of the design with respect to the application (the only link is perhaps the use of REFPROP). The designed turbine could be used in any Organic Rankine cycle application requiring a radial turbine. The reference to the OTEC system is maybe to justify the submission of the paper in this Journal. However, with the current form of the title, the reader expects to read about the cooperation of the designed turbine with a Rankine cycle and furthermore in the context of an OTEC system or how the fact that the turbine will be used in such a system will affect its design or how the improvement of the turbine affects the efficiency of the whole system. Without proposing it as a title, just for the sake of understanding, for example something like ‘… to be used in a Rankine cycle Operating with an OTEC System’ would be more representative of what the paper presents.
  • A lot of formulas are given in the paper. They should, since they describe the methodology. However, some references or some more hints describing their derivation should be given. For example, how are equations (4) or (6) derived? Examine the case to provide some references or an appendix or even a set of sentences describing it derivation is required.
  • Throughout the text, the reader reads ‘Equation (18)’ as well as ‘Equation 19’. Please check the text and provide a uniform presentation, i.e. with or without parentheses whenever you refer to an Equation.
  • Section 3, i.e. Response surface optimization is just mentioned in the paper. Furthermore, a section is dedicated to it for the sake of completeness. However, this section says almost nothing. It provides no information on the formulation of the optimization problem (i.e. its objective function, its constraints). In addition, the method used for its solution has to be described, along with appropriate references. There is just Equation 42 providing no information on how the method was implemented. It is like the implementation of the Response Surface method is meant to be known to the reader. Even it is so, this has to be justified with appropriate references. The very last conclusion of the paper writes: ‘Response surface methodology (RSM) integrated with the 3D-RANS calculation is an effective, reliable, and robust method of the design and optimization of turbo-machinery’. However, the reader is not aware of how this integration was realized!
  • It is in no case clear to this reviewer how the results presented in Figures 8 and 9-15 were obtained. What was the procedure followed? Table 2 mentions ‘design of experiments’. For example, each design variable was in a range and some values assigned to it? At the same time the rest of the design variables were kept constant or all variables were allowed to take different values together? How was the sensitivity analysis carried out? What runs were realized? All these issues are not linked and there is no sequence. The reader is left to fill the gaps by himself! The text refers to ‘central composite design’. It is mentioned in 2-3 points of the text either as ‘central composite design’ or ‘CCD’. What is it? The text is not coherent.
  • The title of the first column in Table 3 is actually valid for some of its rows. The three last lines are not design variable, they are response variables according to the of the paper. In addition, it is proposed that a new fourth column should be added and commented in this Table, providing the percentage (%) difference between the values of the second and third columns.
  • Table 4 is given but no reference of it is made in the text! It is not presented or discussed. Maybe in section 5.1.4. it is Table 4 instead of Table 2.
  • The Nomenclature section is incomplete. There are many symbols used throughout the text that are missing from it. The authors should check it.
  • Finally, are the keywords appropriate? (for example ‘genetic algorithm’?).
  • Conerning Figure 19 the text writes: ‘Figure 19 shows the off-design maps of the turbine. Off design maps were generated to ensure the behavior of the turbine during the off-design conditions. Off-design maps showed that the behavior of the off-design conditions is appropriate and provided additional validity of the adopted process.’ The last sentence should be justified or at least commented.

Author Response

Response to the reviewer 3

Comments

Response

Added on

The authors have conducted an interesting research work. They have described their motivation for their in the introduction and then the implementation a design optimization method for radial turbines. This is based on 1D meanline preliminary design, as well as in CFD and an optimization method. Evaluation of the optimized design is also made by means of 3D Navier-Stokes numerical simulations. The whole idea is good and the research topic modern and interesting. However, this reviewer judges that major revision is required in order the paper to be published. In particular, improvements are required that concern the adequate description of the method used and the clear presentation of results. Relevant remarks and details are given below.

Thank you for reviewing the paper in detail and pointing out the shortcomings that helped in improving the manuscript. All the points observed by the reviewers have been understood completely and are addressed as follows. 

1

The title of the paper is not adequate; it is rather misleading. It actually concerns ‘Design and Optimization of a Radial Turbine’. The second part of the title (‘… for a Rankine Cycle Operating with Ocean Thermal Conversion System’) is just the motivation for this work. Apart from the literature review in the introduction, no other reference is made in the paper to the issues mentioned in the second part of the title. There is no link of the design of the turbine with the Rankine cycle or justification of the design with respect to the application (the only link is perhaps the use of REFPROP). The designed turbine could be used in any Organic Rankine cycle application requiring a radial turbine. The reference to the OTEC system is maybe to justify the submission of the paper in this Journal. However, with the current form of the title, the reader expects to read about the cooperation of the designed turbine with a Rankine cycle and furthermore in the context of an OTEC system or how the fact that the turbine will be used in such a system will affect its design or how the improvement of the turbine affects the efficiency of the whole system. Without proposing it as a title, just for the sake of understanding, for example something like ‘… to be used in a Rankine cycle Operating with an OTEC System’ would be more representative of what the paper presents.

Revivers has raised a valid concern here. The title of the paper has been modified as suggested. The modified title is “Design and Optimization of a Radial Turbine to be used in a Rankine Cycle Operating with an OTEC System’.

2

A lot of formulas are given in the paper. They should, since they describe the methodology. However, some references or some more hints describing their derivation should be given. For example, how are equations (4) or (6) derived? Examine the case to provide some references or an appendix or even a set of sentences describing it derivation is required.

·          

Definition of the has been provided . All the terms involved in Equation (6) are given in the Equations (7-9). The missing derivations are straight forward. However, for further clarity.  references have been provided in the text i.e., [1–3]. Added details have been highlighted in the text.

Page 4 and 5

3

Throughout the text, the reader reads ‘Equation (18)’ as well as ‘Equation 19’. Please check the text and provide a uniform presentation, i.e. with or without parentheses whenever you refer to an Equation

The whole manuscript has been revised and a uniform presentation with parentheses has been adopted. All modification made has been highlighted as well.

4

Section 3, i.e. Response surface optimization is just mentioned in the paper. Furthermore, a section is dedicated to it for the sake of completeness. However, this section says almost nothing. It provides no information on the formulation of the optimization problem (i.e. its objective function, its constraints). In addition, the method used for its solution has to be described, along with appropriate references. There is just Equation 42 providing no information on how the method was implemented. It is like the implementation of the Response Surface method is meant to be known to the reader. Even it is so, this has to be justified with appropriate references. The very last conclusion of the paper writes: ‘Response surface methodology (RSM) integrated with the 3D-RANS calculation is an effective, reliable, and robust method of the design and optimization of turbo-machinery’. However, the reader is not aware of how this integration was realized!

Response surface methodology (RSM) [22] is well capable of the optimization processes when the response is affected by several design variables due to its accuracy, robustness, fewer numerical cost, and simplicity in implantation [23]. A number of studies are presented in the literature where response surface methodology (RSM)  has been chosen for the optimization process when numerous design variables are engaged [24]. Madsen et al.[25,26] employed RSM for the optimization of a diffuser. On the other hand, Ahn et al. [26] make use of it for the optimization of airfoil design. Built on the above debate RSM was implemented for the present analysis to improve the rotor geometry for the parameters given in Table 2.

The relation between the response variable (power) and design factors was chosen using a second-degree model given by Eq. (26)

In the above equation  and  correspond to design variables and responses respectively. Whereas,, and  are regression coefficients. Moreover, term  corresponds to the total number of design variables (e.g. for this study; that are listed in the Table. 2. 

For the design of experiments, the central composite design (CCD) option was used. This option enables the calculation of the overall trends of the Meta-model based on a screening set to better direct the choice of options in Optimal Space-Filling Design. This option could handle up to 20 different design variables. The central composite design (CCD) contains an embedded factorial or fractional factorial design with center points that are augmented with a group of 'star points' that allow estimation of curvature [22]. If the distance from the center of the design space to a factorial point is ±1 unit for each factor, the distance from the center of the design space to a star point is|α| > 1. The precise value of α depends on certain properties desired for the design and on the number of factors involved. To maintain rotate ability, the value of α depends on the number of experimental runs in the factorial portion of the central composite design:

Further details on the response surface methodology and central composite design can be found in the literature [22]. The design variable considered in the current study with defined upper and lower limits is listed in Table 2. The design of the experiment calculated using central composite design (CCD) is listed in Table 3. It should be noted here that each combination of the design variables listed in Table 3 represents a distinct design of the turbine. The design of the turbine for each set of parameters listed in Table 3 was computed using the mean line turbine design code (Figure 5). Now computed power of the turbine as a response variable f(x) for all designs listed in Table 3 can be used to solve the Eq. (26) through regression analysis.

5

It is in no case clear to this reviewer how the results presented in Figures 8 and 9-15 were obtained. What was the procedure followed? Table 2 mentions ‘design of experiments. For example, each design variable was in a range and some values assigned to it? At the same time the rest of the design variables were kept constant or all variables were allowed to take different values together? How was the sensitivity analysis carried out? What runs were realized? All these issues are not linked and there is no sequence. The reader is left to fill the gaps by himself! The text refers to ‘central composite design’. It is mentioned in 2-3 points of the text either as ‘central composite design’ or ‘CCD’. What is it? The text is not coherent.

It has been explained and highlighted in the text that all results presented in figures (8-15) are based on the solved regression model given by Equation (42). Each of the figures (9-15) show variation in the values of the two design variables to examine their effect on the response variable. It should be noted here while the response of the two variables was studied other design variables were kept constant.

For the design of experiments, a central composite design (CCD) option was used. This option enables the calculation of the overall trends of Meta-model based on a screening set to better direct the choice of options in Optimal Space-Filling Design. This option could handle up to 20 different design variables. The central composite design (CCD) contains an embedded factorial or fractional factorial design with center points that are augmented with a group of 'star points' that allow estimation of curvature [4].

The above information has been added and highlighted in the manuscript. An effort has been made to link all the steps involved to conduct the current study.

Pages 9-12

6

he title of the first column in Table 3 is actually valid for some of its rows. The three last lines are not design variable, they are response variables according to the of the paper. In addition, it is proposed that a new fourth column should be added and commented in this Table, providing the percentage (%) difference between the values of the second and third columns.

In Table 4 (previously listed as Table 3) title of the first column has been updated as “Turbine’s geometry and performance parameters” for clarity as advised by the reviewer. Further, column 4 has been added as suggested.

Table 4, Page 17

7

Table 4 is given but no reference of it is made in the text! It is not presented or discussed. Maybe in section 5.1.4. it is Table 4 instead of Table 2.

·          

Citation for Table 5 (previously Table 4) has been added.

8

The Nomenclature section is incomplete. There are many symbols used throughout the text that are missing from it. The authors should check it.

Nomenclature has been upgraded and additions have been highlighted in the text.

9

Finally, are the keywords appropriate? (for example, ‘genetic algorithm’?).

Keywords have been updated. The genetic algorithm has been removed from the keywords list.

10

Concerning Figure 19 the text writes: ‘Figure 19 shows the off-design maps of the turbine. Off design, maps were generated to ensure the behavior of the turbine during the off-design conditions. Off-design maps showed that the behavior of the off-design conditions is appropriate and provided additional validity of the adopted process.’ The last sentence should be justified or at least commented.

The following explanation has been added and highlighted in the text

Figure 19 shows the off-design maps of the turbine. Off design, maps were generated to ensure the behavior of the turbine during the off-design conditions. Off-design maps showed that the behavior of the off-design conditions is appropriate and provided additional validation of the adopted process. This is based on the fact that the operating margin of the optimized turbine in the off-design conditions is sufficiently broad. It can be depicted from Figure 19 that an optimized turbine can be exhibiting reasonable performance by altering the flow rate of the turbine by  of its design point value.

Page 18

References

[1]      R.H. Aungier, Mean Streamline Aerodynamic Performance Analysis of Centrifugal Compressors, J. Turbomach. 117 (1995) 360.

[2]      R.H. Aungier, Turbine Aerodynamics: Axial-Flow and Radial-Flow Turbine Design and Analysis, ASME, Three Park Avenue New York, NY 10016-5990, 2006.

[3]      H. Moustapha, Axial and radial turbines, 1st ed., Concepts NREC, 2003.

[4]      N. Bradley, A. Yi Cheng, Z. Guan Dana Vrajitoru, The response surface methodology, Department of Mathematical Sciences, Indiana University of South Bend, 2007.

[5]      B.S. Mohammed, K.M. Anwar Hossain, J. Ting, E. Swee, G. Wong, M. Abdullahi, Properties of crumb rubber hollow concrete block, J. Clean. Prod. 23 (2012) 57–67.

[6]      B.S. Mohammed, V.C. Khed, M.F. Nuruddin, Rubbercrete mixture optimization using response surface methodology, J. Clean. Prod. 171 (2018) 1605–1621.

[7]      J.I. Madsen, W. Shyy, R.T. Haftka, Response Surface Techniques for Diffuser Shape Optimization, AIAA J. 38 (2000) 1512–1518.

[8]      J. Ahn, H.-J. Kim, D.-H. Lee, O.-H. Rho, Response surface method for airfoil design in transonic flow, J. Aircr. 38 (2001) 231–238.

Round 2

Reviewer 3 Report

The revised version of the paper is much improved. I suggest that it can now be published. They can find attached some scanned pages of their manuscript, where some minor corrections have been highlighted by this reviewer. The authors could make these corrections if they agree. However, this reviewer does not feel he has to see the paper again. It is OK for publication.
